# Correlation between Fecal Calprotectin Levels in Meconium and Vitamin D Levels in Cord Blood: Association with Intestinal Distress

**DOI:** 10.3390/jcm9124089

**Published:** 2020-12-18

**Authors:** Jae Hoon Jung, Sook Hyun Park

**Affiliations:** 1Department of Pediatrics, School of Medicine, Kyunpook National University, Daegu 41404, Korea; wogns7602@gmail.com; 2Division of Neonatology, Kyunpook National University Chilgok Hospital, Daegu 41404, Korea

**Keywords:** vitamin D, calprotectin, necrotizing enterocolitis

## Abstract

We aimed to investigate the correlation between vitamin D status in cord blood and fecal calprotectin concentrations in meconium, and also find their association with intestinal distress symptoms during the first two weeks of life. Two hundred and twenty-eight newborns were enrolled in the study who were delivered at Kyungpook National University Children’s Hospital between July 2016 and August 2017. The first passed meconium samples were collected for fecal calprotectin analysis. Intestinal distress involved infants with necrotizing enterocolitis (NEC) and other feeding interruption signs. The median gestational age of the population was 37.0 (34.3–38.4) weeks, and the median birth weight was 2635 (2100–3268) g. The median fecal calprotectin levels in meconium were 134.1 (55.6–403.2) μg/g (range: 11.5–2000 μg/g) and the median 25-hydroxyvitamin D (25-OHD) concentrations in cord blood were 21.0 (15.5–28.8) ng/mL. Sixty infants (26.3%) had intestinal distress, including four patients (1.8%) diagnosed as having NEC. Higher fecal calprotectin concentrations (398.2 (131.8–900.2) μg/g vs. 105.6 (39.4–248.5) μg/g, *p* < 0.001) and lower 25-OHD levels (17.9 (12.8–22.1) ng/mL vs. 23.2 (17.2–33.0) ng/mL, *p* < 0.001) were found in infants with intestinal distress compared to infants without intestinal distress. The cut-off value was set at 359.8 μg/g with a sensitivity of 0.53 and a specificity of 0.82 for the development of intestinal distress in the first two weeks of life. Serum 25-OHD levels in cord blood were inversely correlated with fecal calprotectin concentrations in meconium.

## 1. Introduction

Vitamin D is a major regulator of immune function and inflammation and has modulatory effects on the cells of the adaptive and innate immune system. Vitamin D provided biological effects through vitamin D receptors expressed in the kidney, colonic mucosa, and immune cells [1,2,3]. Vitamin D modulates innate immunity by regulating antimicrobial peptides, which have functions including microbiocidal activity and chemotaxis of inflammatory immune cells [4,5]. Consequently, vitamin D deficiency can result in a pro-inflammatory state and increase susceptibility to intestinal infections [6]. Vitamin D deficiency in the gastrointestinal tract has been presumed to decrease colonic bacterial clearance and lead to expressions of tight junctions in the intestinal epithelium and elevated Th1-mediated inflammation [7,8,9]. Vitamin D levels have been reported to be associated with severity and the activity of inflammatory bowel disease in adolescents and adults [10]. In infants, maternal 25-hydroxyvitamin D (25-OHD) levels have been associated with necrotizing enterocolitis (NEC) development [11].

Fecal calprotectin is derived mainly from activated neutrophils that have migrated through the gastrointestinal mucosa [12,13]. Migration and accumulation in transepithelial neutrophil at the mucosal surface and within the intestinal lumen are a hallmark of digestive inflammatory pathology [14]. Studies have reported that increased fecal calprotectin levels are correlated with an increased turnover of leukocytes in the intestinal epithelial cells and granulocyte migration toward intestinal lumen [12,15]. It has been previously investigated as a biomarker of mucosal inflammation in various conditions such as NEC, inflammatory bowel disease, and intestinal ischemia-reperfusion [16,17,18,19,20,21]. Recently, fecal calprotectin levels have been reported to predict cow’s milk protein allergy and atopic disease [22,23]. Increased calprotectin levels have also been observed in cases of intolerance to enteral feeding [24].

Under the hypothesis that fecal calprotectin in meconium may reflect the inflammatory status in fetal intestines, the present study aimed to investigate the correlation between vitamin D levels in cord blood and fecal calprotectin concentrations in meconium and also explore its association and intestinal distress symptoms during the first two weeks of life.

## 2. Materials and Methods

### 2.1. Study Population and Ethics

The study group enrolled 228 infants who were delivered at Kyungpook National University Children’s Hospital between July 2016 and August 2017. Only patients who had results of 25-hydroxyvitamin D (25-OHD) levels through cord blood or at birth were included. The first stool after birth was collected for fecal calprotectin analysis in the meconium. The study excluded patients with any chromosomal or major congenital anomalies. The current study was approved by the Institutional Review Board of Kyungpook National University Chilgok Hospital and was given a waiver of informed consent (2017-03-023).

### 2.2. Maternal and Neonatal Demographic Data

Maternal and neonatal demographic data were collected through medical record reviews. The maternal demographic features that were recorded included maternal age, premature rupture of membranes (PROM), diabetes, and pregnancy-induced hypertension (PIH). They also included maternal use of antibiotics and steroids for one week before delivery. Neonatal clinical data included gestational age, birth weight, sex, delivery mode, Apgar score at 1 min and 5 min, and small-for-gestational age (SGA) which was defined as under the 10-percentile range for gestational age using the 2013 Fenton growth calculator for preterm infants [25].

### 2.3. Measurement of Fecal Calprotectin and 25-OHD in Cord Blood

All fecal samples were frozen and stored at −20 °C immediately after collection. Analysis took an average of two to three days. Fecal samples were measured for calprotectin concentration using the supernatant of a centrifuged solution from 0.2 g of stool that was used to evaluate the calprotectin level by ImmunoCap (Phadia 250, Phadia AB (Thermo Scientific), Uppsala, Sweden) after using a Fluorescence Enzyme Immunoassay (FEIA). The detected range of fecal calprotectin concentrations was 11.5–2000 μg/g. The serum 25-OHD levels were measured using radioimmunoassay (DIAsource 25OH-Vit.D3-Ria-CT kit, DIAsource Immuno-Assays S.A., Louvain-la-Neuve, Belgium).

### 2.4. Definition of Intestinal Distress

In the present study, intestinal distress involved NEC and other feeding interruption signs during the first two weeks of postnatal age. NEC was classified according to the modified Bell’s classification [26]. Other feeding interruption signs included infants with one or more episodes of increased gastric residuals (>50%) of previous feeding volume, vomiting, bloody stool, diarrhea, abdominal distension (increased abdominal circumstance > 2 cm), and visible bowel loops on physical examination. Intestinal distress was determined by one neonatologist or two pediatric residents.

### 2.5. Statistical Analysis

Statistical analysis was performed using IBM SPSS Statistics 26.0 software (IBM Corp., Armonk, NY, USA). The Shapiro–Wilk test was applied to check normal distribution between two groups, and the skewness and kurtosis of each variable were studied to test normal distribution. Fecal calprotectin by demographic and clinical factors was compared using the Mann–Whitney test or the chi-square test according to continuous or categorical variables. The Spearman’s rank correlation coefficient was used to analyze the correlation between fecal calprotectin concentrations and 25-OHD levels. The analysis of potential confounding factors affecting intestinal distress were performed using logistic regression. The cut-off value for fecal calprotectin in the prediction of intestinal distress was determined using the receiver operating characteristics curve analysis. Statistical significance was defined as *p* value < 0.05.

## 3. Results

### 3.1. Demographic and Clinical Features of the Study Population

Demographic and clinical characteristics of patients are shown in Table 1. The median gestational age of the population was 37.0 (34.3–38.4) weeks, and the median birth weight was 2635 (2100–3268) g. The median fecal calprotectin levels in meconium were 134.1 (55.6–403.15) μg/g (range: 11.5–2000 μg/g) and the median 25-OHD concentrations in cord blood were 21.0 (15.5–28.8) ng/mL. Sixty infants (26.3%) had intestinal distress, including four patients (1.8%) who were diagnosed as having NEC. The number of infants with maternal medical history of antibiotics and steroid use was 40 (17.5%), and 31 (13.6%), respectively.

### 3.2. Fecal Calprotectin Concentrations by Maternal and Neonatal Factors

Table 2 shows the comparison of fecal calprotectin concentrations by maternal factors. The median fecal calprotectin of infants with maternal PROM (148.1 (62.1–416.6) μg/g) was higher than that of infants without maternal PROM (86.0 (24.9–170.3)) μg/g, *p* = 0.039). There were no statistical differences of fecal calprotectin by maternal PIH or preeclampsia, diabetes, and maternal use of antibiotics and steroid. There were also no significant differences in the median value of fecal calprotectin by sex, prematurity, SGA, or delivery mode (Table 3).

### 3.3. Correlation between Fecal Calprotectin Concentrations and 25-OHD Levels

Fecal calprotectin levels in meconium were not correlated with gestational age (ρ = −0.080, *p* = 0.232) and birth weight (ρ = −0.020, *p* = 0.769). Apgar scores at 1 min (ρ = 0.096, *p* = 0.161) and 5 min (ρ = 0.116, *p* = 0.090) were not correlated with fecal calprotectin levels in meconium. Results also indicate that fecal calprotectin concentrations were significantly correlated with 25-OHD levels (ρ = −0.352, *p* < 0.001, Figure 1).

### 3.4. Association between Intestinal Distress and Fecal Calprotectin Concentrations

Infants with intestinal distress had lower gestational age and lower birth weight, and older maternal age (Table 4). Higher fecal calprotectin concentrations (398.2 (131.8–900.2) μg/g vs. 105.6 (39.4–248.5) μg/g, *p* < 0.001) and lower 25-OHD levels (17.9 (12.8–22.1) ng/mL vs. 23.2 (17.2–33.0) ng/mL, *p* < 0.001) were found in infants with intestinal distress compared to infants without intestinal distress. The fecal calprotectin concentrations of the two infants diagnosed as having NEC stage III were 32.3 μg/g and 86.0 μg/g, respectively, and the two infants with NEC stage IIb were 180.3 μg/g and 169.5 μg/g, respectively. The 25-OHD levels of the two infants diagnosed as having NEC stage III were 13.77 ng/mL and 13.37 ng/mL, respectively, and the two infants with NEC stage IIb were 21.85 ng/mL and 30.63 ng/mL, respectively.

After adjusting confounding factors, intestinal distress was associated with lower gestational age, older maternal age, maternal PIH, and lower Apgar score (Table 5). Lower 25-OHD levels (odds ratio (OR), 0.943; 95% confidence interval (CI), 0.897–0.991, *p* = 0.021) and higher calprotectin concentrations (OR, 1.003; 95% CI, 1.002–1.004, *p* < 0.001) were associated with an increased risk of intestinal distress.

The cut-off value was set at 359.8 μg/g with a sensitivity of 0.53 and a specificity of 0.82 for the development of intestinal distress in the first two weeks of life. The area under the curve value was 0.738 (*p* < 0.001, 95% CI: 0.661–0.815) (Figure 2).

## 4. Discussion

In the present study, median fecal calprotectin levels in meconium were 134.1 (55.6–403.2) μg/g. Fecal calprotectin concentrations in meconium were inversely correlated with 25-OHD levels of cord blood. After adjusting for confounding factors, it was found that intestinal distress was associated with lower gestational age, older maternal age, maternal preeclampsia, lower Apgar score, lower 25-OHD levels, and higher fecal calprotectin levels.

In a previous study, it was shown that fecal calprotectin concentrations in meconium were significantly correlated with birth weight, gestational age, and Apgar score at 5 min [27]. It is thought that prematurity could be associated with immature intestinal mucosa that could induce the transepithelial migration of neutrophil granulocytes or macrophages into the lumen [27]. However, another study similar to the present study showed no statistically significant correlation between fecal calprotectin levels and birth weight, gestational age, or intrauterine growth [28]. This suggests that there might be other factors, aside from prematurity, that may be involved, such as ischemic condition and inflammation determining fecal calprotectin concentrations.

Maternal factors associated with placenta-related oxidative stress can induce endothelial cell injury; this is aggravated in the case of infants having increased mucosal inflammation [29]. It results in a high level of calprotectin in infants with maternal preeclampsia [21]. Under conditions of perinatal asphyxia, high levels of calprotectin in meconium were also found and indicate that ischemic damage to the intestinal mucosa may facilitate the passage of neutrophils [27]. In the present study, Apgar scores at 1 min and 5 min were not correlated with fecal calprotectin levels in meconium, but they were with intestinal distress. Infants with maternal inflammatory conditions, including maternal PROM, had higher fecal calprotectin levels compared to infants without maternal PROM. However, maternal preeclampsia was not correlated with fecal calprotectin levels but, rather, it was associated with intestinal distress.

Vitamin D promotes synthesis of the anti-inflammatory cytokines and inhibits the synthesis of pro-inflammatory cytokines [6]. Active vitamin D is used to increase tight junction proteins, which helps to protect the intestinal mucosal barrier [6,30]. Vitamin D receptor knockout mice have a tendency to cause infection, cancer, and inflammation [31]. Expression and localization of tight junction proteins are inhibited during inflammation, causing intestinal permeability and bacterial translocation [32,33]. Thus, the immature immune system, weak gastrointestinal mucosal barrier, and the abnormal colonization of the intestine predispose it to intestinal necrosis caused by progressive inflammation and bacterial invasion [34,35]. Increasing gut permeability and active immune response promotes transepithelial migration of neutrophils, which induces high fecal calprotectin in intestinal lumen [28]. Vitamin D may play a vital role as an immunomodulator in processing the regulation of migration and accumulation of fecal calprotectin.

The fecal calprotectin concentrations of infants diagnosed as having NEC stage III were reported as 32.3 μg/g, and 86.0 μg/g, respectively. The results were relatively lower than the median value of all infants in the study group, as well as those of infants with intestinal distress. The association between significantly low meconium calprotectin concentrations and developed fulminant NEC in preterm infants was also previously reported [36]. It can therefore be understood that under extremely severe inflammatory conditions, the impaired function of granulocytes in the intestinal lumen causes the decreased migration and release of calprotectin [36]. In the present study, the cut-off value of 359.8 μg/g with a sensitivity of 0.53 and a specificity of 0.82 indicated the development of intestinal distress in the first two weeks of life. This result represents similar fecal calprotectin levels and lower sensitivity than a previous study which had a cut-off value of 363 μg/g with a sensitivity of 0.65 and a specificity of 0.82 set for mild digestive symptoms [37]. Higher gestational age and strict criteria for intestinal distress may lead to the representation of a relatively high proportion of infants with mild distress signs.

To the best of our knowledge, the present study is the first of its kind to present a significant correlation between vitamin D and calprotectin in newborns, which suggests that the role of vitamin D functions as an immune modulator with anti-inflammatory effects. When considered in connection with a previous study found to have low maternal vitamin D levels in NEC infants [11], it is thought that maintaining adequate vitamin D levels during pregnancy may affect intestinal conditions in the early neonatal period. Abnormal intestinal colonization and an undeveloped balance of immune system response may cause feeding intolerance in the early postnatal period [38], which is also a known association with the migration of high fecal calprotectin into the gut lumen. The present study found that fecal calprotectin was significantly higher in infants with intestinal distress signs. Through these results, fecal calprotectin in meconium can be considered to be a predictive factor of feeding intolerance and also of inflammatory changes of the intestine which can cause feeding interruption. In inflammatory bowel disease patients, it remains unclear whether vitamin D deficiency can lead to or result in severe disease activity. Nonetheless, fecal calprotectin in meconium can be used to find accurate correlations between fecal calprotectin and 25-OHD levels, as it is a component that is measured before disease progression.

The present study is observational, and we reviewed all participants belonging to the inclusion criteria, which may affect the disproportion of the study group. We analyzed the distribution of each variable and between groups to minimize statistical flaws. The distribution of fecal calprotectin in meconium was similar to that of a previous study [27]. However, the participants of the present study included more moderate-to-late preterm infants compared with extremely preterm infants, so the incidence of NEC was reported to be low. Therefore, it has a limited ability to evaluate the role of fecal calprotectin as a predictor of NEC. Considering previous reports found that fecal calprotectin tends to vary widely in the first month, the present study only investigated the correlation with intestinal distress in the early neonatal period. The present study suggests that fecal calprotectin in meconium can predict early intestinal distress but requires the further exploration of longitudinal changes according to the time of intestinal signs. The factors affecting fecal calprotectin in infants have not been thoroughly understood and different results have previously been reported on the correlation with gestational age, birth weight, hypoxic condition, and maternal diseases. Further research is necessary to investigate the association between maternal and neonatal demographics, and clinical factors, along with fecal calprotectin levels.

## 5. Conclusions

Infants with intestinal distress had higher fecal calprotectin levels compared with infants without intestinal distress. Serum 25-OHD levels in cord blood were inversely correlated with fecal calprotectin concentrations in meconium, which suggests maintaining appropriate vitamin D status during the fetal period may prevent intestinal distress after birth.

## Figures and Tables

**Figure 1 jcm-09-04089-f001:**
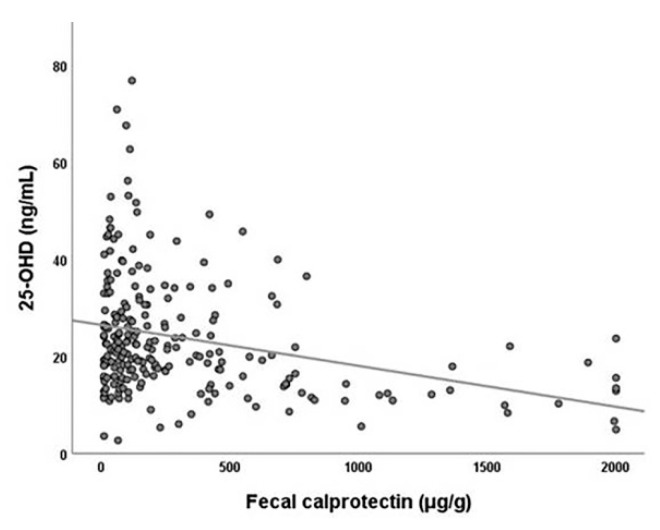
Correlation between 25-hydroxyvitamin D (25-OHD) levels in cord blood and fecal calprotectin concentration of meconium.

**Figure 2 jcm-09-04089-f002:**
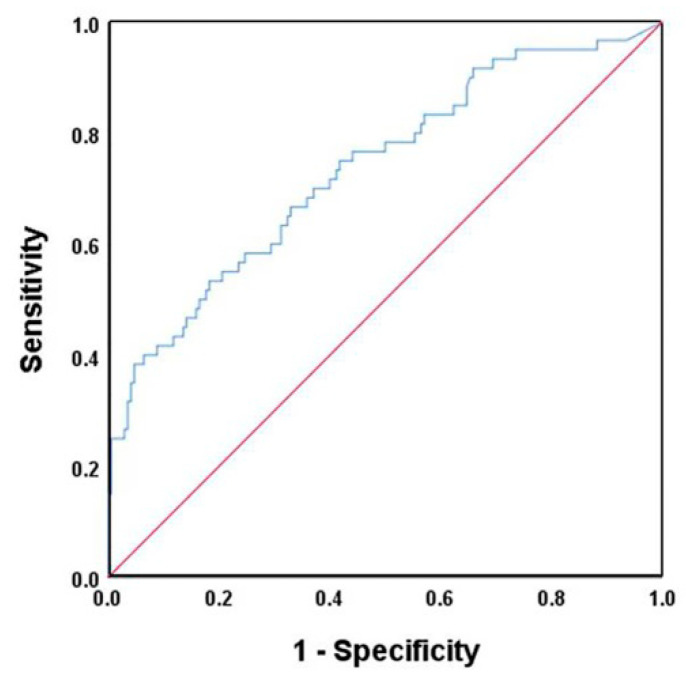
Receiver operating characteristics curves identifying infants with intestinal distress.

**Table 1 jcm-09-04089-t001:** Demographic and clinical characteristics of the population.

Variable	Median (IQR) or *n* (%)
Male sex	118 (51.8)
Gestational age (wks)	37.0 (34.3–38.4)
Birth weight (g)	2635 (2100–3268)
SGA	22 (9.6)
*Delivery mode*	
Vaginal delivery	68 (29.8)
Cesarean section	160 (70.2)
*Maternal features*	
Age	33.0 (30.0–36.0)
PROM	35 (14.8)
PIH or Preeclampsia	24 (10.5)
Diabetes	24 (10.5)
Antibiotics	40 (17.5)
Steroid	31 (13.6)
*Neonatal features*	
Apgar score at 1 min	7.0 (7.0–8.0)
Apgar score at 5 min	9.0 (8.0–9.0)
Intestinal distress	60 (26.3)
NEC	4 (1.8)
25-OHD (ng/mL)	21.0 (15.5–28.8)
Calprotectin (μg/g)	134.1 (55.6–403.2)

Data reported as frequency *n* (%) or median (IQR). Abbreviations: SGA, small-for-gestational age; PROM, premature rupture of membranes; PIH, pregnancy-induced hypertension; NEC, necrotizing enterocolitis, 25-OHD, 25-hydroxyvitamin D.

**Table 2 jcm-09-04089-t002:** Differences of fecal calprotectin concentrations in meconium by maternal factors.

Factors		*n*	FC (μg/g)	*p*-Value
Diabetes	Yes	24	248.6 (84.0–524.0)	0.125
	No	204	126.8 (53.0–403.2)	
PROM	Yes	35	148.1 (62.1–416.6)	0.039
	No	193	86.0 (24.9–170.3)	
PIH or preeclampsia	Yes	24	177.8 (62.3–356.5)	0.739
	No	204	131.7 (55.6–414.3)	
Use of antibiotics	Yes	40	105.8 (49.4–458.3)	0.634
	No	188	134.1 (56.7–369.6)	
Use of steroid	Yes	31	105.8 (68.9–318.9)	0.975
	No	197	136.3 (53.1–416.6)	

Data reported as frequency median (IQR). Abbreviations: FC, fecal calprotectin; PROM, premature rupture of membranes; PIH, pregnancy induced hypertension.

**Table 3 jcm-09-04089-t003:** Differences of fecal calprotectin concentrations in meconium by neonatal factors.

Factors	*n*	FC (μg/g)	*p*-Value
*Sex*			
Female	110	122.4 (53.3–364.5)	0.455
Male	118	148.1 (61.5–420.8)	
*Gestational age*			
Preterm	111	134.4 (61.8–437.2)	0.639
Full-term	117	133.8 (52.8–301.5)	
*Fetal growth*			
SGA	22	92.3 (22.5–191.3)	0.086
AGA	206	137.2 (61.5–415.8)	
*Delivery mode*			
Vaginal delivery	68	141.4 (77.6–397.4)	0.420
Cesarean section	160	123.7 (52.9–403.2)	

Data reported as frequency median (IQR). Abbreviations: FC, fecal calprotectin; SGA, small-for-gestational age; AGA, appropriate-for-gestational age.

**Table 4 jcm-09-04089-t004:** Demographic and clinical characteristics of patients with intestinal distress.

Variable	ID (*n* = 60)	Non-ID (*n* = 168)	*p*-Value
Male sex	32 (53.3)	86 (51.2)	0.880
Gestational age (weeks)	34.5 (32.1–37.1)	37.4 (35.1–38.7)	<0.001
Birth weight (g)	2185 (1663–2683)	2805 (2238–3373)	<0.001
SGA	4 (6.7)	18 (10.7)	0.452
Cesarean section	46 (76.7)	114 (67.9)	0.250
*Maternal features*			
Age	34.0 (30.0–37.0)	32.0 (29.0–35.0)	0.034
PROM	11 (18.3)	24 (14.3)	0.531
PIH or preeclampsia	8 (13.3)	16 (9.5)	0.463
Diabetes	9 (15.0)	15 (8.9)	0.221
Antibiotics	16 (26.6)	24 (14.3)	0.263
Steroid	12 (20.0)	19 (11.3)	0.123
*Neonatal features*			
Apgar score at 1 min	7.0 (6.0–8.0)	7.0 (7.0–8.0)	0.327
Apgar score at 5 min	9.0 (8.0–9.0)	9.0 (8.0–9.0)	0.663
25-OHD (ng/mL)	17.9 (12.8–22.1)	23.2 (17.2–33.0)	<0.001
Calprotectin (μg/g)	398.2 (131.8–900.2)	105.6 (39.4–248.5)	<0.001

Data reported as frequency *n* (%) or median (IQR). Abbreviations: ID, intestinal distress; SGA, small-for-gestational age; PROM, premature rupture of membranes; PIH, pregnancy induced hypertension; NEC, necrotizing enterocolitis; 25-OHD, 25-hydroxyvitamin D.

**Table 5 jcm-09-04089-t005:** Logistic regression analysis of associated factors with intestinal distress.

Factors	Adjusted		
	Odds Ratio	95% CI	*p*-Value
Male sex	0.673	0.278–1.630	0.380
Gestational age (weeks)	0.675	0.471–0.968	0.032
Birth weight (g)	0.999	0.998–1.001	0.448
SGA	1.828	0.274–12.217	0.533
Cesarean section	0.558	0.199–1.565	0.267
*Maternal features*			
Age	1.164	1.049–1.292	0.004
PROM	1.541	0.389–6.097	0.538
PIH or preeclampsia	5.240	1.204–22.803	0.027
Diabetes	1.801	0.471–6.882	0.390
Antibiotics	3.649	0.577–23.098	0.169
Steroid	1.515	0.388–5.910	0.550
*Neonatal features*			
Apgar score at 1 min	0.517	0.271–0.986	0.045
Apgar score at 5 min	0.344	0.125–0.951	0.040
25-OHD (ng/mL)	0.943	0.897–0.991	0.021
Calprotectin (μg/g)	1.003	1.002–1.004	<0.001

Abbreviations: CI, Confidence interval; ID, intestinal distress; SGA, small-for-gestational age; PROM, premature rupture of membranes; PIH, pregnancy induced hypertension; NEC, necrotizing enterocolitis; 25-OHD, 25-hydroxyvitamin D.

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
