# Peer review of "Correlation between Fecal Calprotectin Levels in Meconium and Vitamin D Levels in Cord Blood: Association with Intestinal Distress"

_jcm, 2020, doi:10.3390/jcm9124089_

Round 1

Reviewer 1 Report

The manuscript submitted for review is an observational study. I believe it is suitable for publication without corrections. The presented topic, related to the elements of the intestinal barrier tightness and the concentration of vitamin D3, is currently of interest to scientists. Both the tightness of the intestinal barrier and the level of vitamin D3 in the body are believed to be important for immune responses. The topic is important and the research undertaken can be considered innovative. The concentration of calprotectin in meconium of newborns was investigated in correlation with the concentration of vitamin D3 in umbilical cord blood. The relationships between the concentrations of the studied biomarkers in various clinical situations were searched for. All elements of the scientific work were properly presented, material and methods were well described. The statistical methods were selected appropriately, the results are presented in a very clear way in tables and graphs. The discussion is rich, and the literature is properly selected. The conclusions correspond with the assumed goals.  

Author Response

I really appreciate your comment and decision. Thank you very much.

Reviewer 2 Report

In the current manuscript, the authors prospectively determined meconium calprotectin levels, and correlated this to cord blood 25-OHD levels and clinical status in terms of intestinal distress. This study is well designed and supports the notion that vitamin D levels may be important during the fetal period to prevent immune dysregulation and associated clinical consequences in the gut.

A few minor comments:

Line 43: Do the authors mean: “transepithelial neutrophils” ?

Lines 58:61: Please specify in more details how the patient selection process occurred; which patients get cord blood 25-OHD levels tested in this hospital? For what reason?

Lines 82-86: Please elaborate on the definition of intestinal distress; how was this determined? When was the intestinal distress evaluated (how long after birth, until when), and by whom?

Line 86: Do the authors mean visible bowel loops on abdominal x-ray, or physical exam?

Lines 101-103: At what time/age did the patients experience intestinal distress?

Lines 136-138: Could the authors report the 25-OHD levels of the patients experiencing NEC?

Lines 143 and 160: Do the authors mean “confounding factors”?

Line 162: The dot is missing at the end of the sentence

Line 178: The patients in whom maternal gestational diabetes was present in fact had higher levels of fecal calprotectin in the author’s study

Line 187: Replace “cuased” by “caused”

Author Response

Thank you for your comment and decision. We thoroughly reviewed your comment, and I have revised what you mentioned as possible as I could. Thank you.

Line 43: Do the authors mean: “transepithelial neutrophils” ?

Thank you for your sincere review and comment. We’ve corrected the order of words.

Lines 58:61: Please specify in more details how the patient selection process occurred; which patients get cord blood 25-OHD levels tested in this hospital? For what reason?

Thank you for your comment to give us a chance to explain why we had tested 25-OHD levels. In our unit, in case that the baby admits NICU after delivery, the result of blood samples is usually analyzed by their cord blood. We screen the 25-OHD levels of all admitted infants if it’s possible, to evaluate and manage vitamin D deficiency. It’s based on our previous study, which indicated that preterm infants in Korean had higher prevalence of vitamin D deficiency compared to other countries’ reports.

Lines 82-86: Please elaborate on the definition of intestinal distress; how was this determined? When was the intestinal distress evaluated (how long after birth, until when), and by whom?Lines 101-103: At what time/age did the patients experience intestinal distress?

Thank you for your comment. We defined as the inclusion criteria that infants had intestinal distress signs during the first 2 weeks of postnatal age. Intestinal distress was evaluated by physician (neonatologist or pediatric residents). We have added the content (line 81-87).

Line 86: Do the authors mean visible bowel loops on abdominal x-ray, or physical exam?

Thank you for your comment to give us a chance to verify the definition. It means on physical exam, so we’ve added the content (line 86-87).

Lines 136-138: Could the authors report the 25-OHD levels of the patients experiencing NEC?

Thank you for your comment. We’ve added the 25-OHD levels (line 143-145).

Lines 143 and 160: Do the authors mean “confounding factors”?

Thank you for your sincere review. We’ve corrected what you mentioned.

Line 162: The dot is missing at the end of the sentence

Thank you for your sincere review. We’ve corrected what you mentioned.

Line 178: The patients in whom maternal gestational diabetes was present in fact had higher levels of fecal calprotectin in the author’s study

Thank you for your comment. We analyzed the fecal calprotectin between groups, but the result of reperformed statistics, the difference between infants with and without maternal diabetes was not significant (revised table 2).

Line 187: Replace “cuased” by “caused”

Thank you for your sincere review. We’ve corrected what you mentioned.

Reviewer 3 Report

Jae Hoon Jung and Sook-Hyun Park in their article entitled: ”Correlation Between Fecal Calprotectin Levels in Meconium and Vitamin D Levels in Cord Blood: Association with Intestinal Distress” tried to assess the correlation between vitamin D and FC among infants.

I have few major concerns about this article:

(1).

Considering the results of the study, it appears that the distribution for most of the variables was not normal. In many of the results, the standard deviation significantly (e.g., 150%) exceeds the average result. This shows that the distribution was not normal, and this further determines the selection of the appropriate tests. If the selection of tests was not preceded by a distribution evaluation, it should be assumed that the type of statistical tests was not well selected. Moreover, the Authors give "mean" once, and sometimes "medians", and simultaneously provide SD, not quartiles. Mean and median is not the same value.

(2).

Another problem is the selection of the study groups. First, there is too much disproportion between the study groups, especially since the intestinal group is much smaller than controls. Second, within the group with intestinal distress we have patients with various etiological causes. It should not be like that. There is an obvious difference between patients with NEC and patients with for example functional bowel disorders. They cannot be assessed together in one group.

(3).

There is no convincing explanation for the linkage of vitamin D3 with FC. Low vitamin D3 levels are very common in various populations, and it is a bold assumption to say that this translates into, for example, NEC.

Author Response

Thank you for your comment and decision. We thoroughly reviewed your comment, and I have revised what you mentioned as possible as I could. Thank you.

(1). Considering the results of the study, it appears that the distribution for most of the variables was not normal. In many of the results, the standard deviation significantly (e.g., 150%) exceeds the average result. This shows that the distribution was not normal, and this further determines the selection of the appropriate tests. If the selection of tests was not preceded by a distribution evaluation, it should be assumed that the type of statistical tests was not well selected. Moreover, the Authors give "mean" once, and sometimes "medians", and simultaneously provide SD, not quartiles. Mean and median is not the same value.

 Thank you for your comment. We did not analyze the data properly by the distribution. The statistical analysis has been totally reviewed and reperformed. We checked the normality of each variable and between groups (Line 89-92). The results were revised mean/SD to median/IQR, which were performed by Mann-Whitney test; because variables between groups did not satisfy with normal distribution (Table 2,3, and 4)

(2).Another problem is the selection of the study groups. First, there is too much disproportion between the study groups, especially since the intestinal group is much smaller than controls. Second, within the group with intestinal distress we have patients with various etiological causes. It should not be like that. There is an obvious difference between patients with NEC and patients with for example functional bowel disorders. They cannot be assessed together in one group.

(1) Thank you very much for your comment. The present study is an observational study, therefore we did not match the number of between groups. We tested normal distribution for comparison between two group to overcome the discrepancy of numbers. It also was analyzed by Mann-Whitney test to compare both groups according to the distribution. We’ve added this limitation in discussion (Line 228-230).

(2) Thank you for your comment. The purpose of this study was to determine the correlation between fecal calprotectin and intestinal distress /fecal calprotectin and vitamin D under the assumption that fecal calprotectin is one of the bowel inflammation markers. In present study, the conditions that caused intestinal distress were very diverse, which means that the rise of fecal calprotectin can occur due to various causes. The purpose of our study was not to investigate the cause of the rise of fecal calprotectin (including NEC) or the change of fecal calprotectin according to various causes. We aimed to find out the correlation between higher concentrations of fecal calprotectin in meconium, which was hypothesized to reflect the degree of inflammation during fetal period, and intestinal distress during the early period of life. So, we chose regression analysis to analyze confounding factors that may affect fecal calprotectin in meconium.

(3).There is no convincing explanation for the linkage of vitamin D3 with FC. Low vitamin D3 levels are very common in various populations, and it is a bold assumption to say that this translates into, for example, NEC.

Thank you for your comment. The roles of vitamin D as an anti-inflammatory effect and protecting the intestinal epithelial barrier can influence immunity and microbiota in intestines, and by these mechanisms, the correlation between fecal calprotectin and vitamin D has been reported in several studies in adult IBD patients (Reference 6, 8, 10). In addition, studies on the role of fecal calprotectin as a predictive value in NEC in neonates have been reported (Reference 16, 19, 21, 37), and one study on the correlation between vitamin D deficiency and NEC occurrence have been reported (Reference 11). Clinically, correlations between vitamin D and fecal calprotectin have been reported, but it’s still unknown the mechanism how vitamin D directly affects fecal calprotectin. Therefore, we assumed that vitamin D as an immune modulator and anti-inflammatory effect and its role in protecting the intestinal barrier can affect the intra-luminal secretion of fecal calprotein.

Round 2

Reviewer 3 Report

The authors made corrections to the article but they did not respond to all my comments.

(1).

In my comments - I paid attention to incorrectly selected statistical methods. The present version of the article still lacks this scope. If the data is not normally distributed, no mean values ​​should be given at all, as they do not reflect the essence of the measurement. Apart from the medians, the article also includes means of the same data.

Moreover, correlations in the article were tested using Pearson’s coefficient, what should not be done in the situation of lacking the normal distribution. You should use Spearman coef.

(2).

Another problem was the selection of the study groups. In my opinion, the study group cannot be so broadly selected. It is like, for example, studying the correlation between the level of “ALT” and “abdominal pain” in one study group including patients with cholecystolithiasis, appendicitis and IBS.

Author Response

Thank you for your comment. We are very sorry that there are still amendments we have not responded appropriately, even we tried our best to revise as your comments. We have fully reviewed and discussed your important comments, we have revised and answered them as possible as we could. Thank you very much.

(1).

In my comments - I paid attention to incorrectly selected statistical methods. The present version of the article still lacks this scope. If the data is not normally distributed, no mean values ​​should be given at all, as they do not reflect the essence of the measurement. Apart from the medians, the article also includes means of the same data.

Moreover, correlations in the article were tested using Pearson’s coefficient, what should not be done in the situation of lacking the normal distribution. You should use Spearman coef.

Thank you for your comment. Our data had been reviewed and analyzed again by the team of statisticians before we submit our first revised manuscript. We had checked the skewness and kurtosis of all data to prove normal distribution. The skewness of 25-OHD levels (1.34) and fecal calprotectin levels (2.38) was below 3, and kurtosis of both levels (25-OHD: 2.44, fecal calprotectin: 5.52) was below 8 (or 10), which indicated normal distribution according to the reference, so we followed their advice. But now we’ve checked the normal distribution again with Shapiro-Wilk test. We changed the method of correlation test (Line 92, 127-131). We have revised all data from mean/SD to median (IQR) including Table 1 and in all contents (Line 16-21, 100-104, 111-113, 115, 166, 201) as you mentioned. Thank you again for your sincere advice.

(2).

Another problem was the selection of the study groups. In my opinion, the study group cannot be so broadly selected. It is like, for example, studying the correlation between the level of “ALT” and “abdominal pain” in one study group including patients with cholecystolithiasis, appendicitis and IBS.

Thank you for your comment. At first, our object was to find out the correlation between fecal calprotectin level in meconium and 25-OHD levels in cord blood. And also we wanted to show correlation with clinical manifestation and those levels assumed indicating an inflammatory state during fetal period. In present study, several diseases such as cow milk protein allergy, NEC, and feeding intolerance associated with infectious condition were included by clinical symptom, not according to a disease entity as you mentioned it. We totally agree with your opinion. We also had difficulties to determine the inclusion criteria in newborn infants who had particularly very vague symptoms, in case of NEC, sepsis or allergy, they can present only systemic symptom, or only feeding intolerance. Therefore, we designed that we collected as intestinal distress conditions; focused on known or reported conditions affecting or affected by fecal calprotectin in neonates (based on previous studies; reference 16-19,21,22,24,37), and then aimed to find out the correlation with fecal calprotectin concentrations (25-OHD levels) and intestinal distress signs in our study group.